# Study on the Driver Visual Workload in High-Density Interchange-Merging Areas Based on a Field Driving Test

**DOI:** 10.3390/s24196247

**Published:** 2024-09-26

**Authors:** Yue Zhang, Pei Jiang, Siqi Wang, Shuang Cheng, Jin Xu, Yawen Liu

**Affiliations:** 1College of Traffic & Transportation, Chongqing Jiaotong University, Chongqing 400074, China; zyyyyyy1007@163.com; 2Shenzhen General Integrated Transportation and Municipal Engineering Design & Research Institute Co., Ltd., Shenzhen 518003, China; jp1986cn@126.com (P.J.); 15989300415@163.com (S.C.); liuyawen2016@163.com (Y.L.); 3Changsha BYD Automobile Co., Ltd., Changsha 410019, China; 18832011843@163.com

**Keywords:** road engineering, high-density interchange, eye movement, visual workload, driving behavior, traffic safety

## Abstract

A visual workload model was constructed to determine and evaluate drivers’ visual workload characteristics in high-density interchange-merging areas. Five interchanges were selected, and a real-vehicle driving test was conducted with 47 participants. To address the differences in drivers’ visual characteristics in the interchange cluster merging areas, the Criteria Importance Through Intercriteria Correlation (CRITIC) objective weighting method was employed. Six visual parameters were selected to establish a comprehensive evaluation model for the visual workload in high-density interchange-merging areas. The results show that the average scanning frequency and average pupil area change rate are most strongly correlated with the visual workload, whereas the average duration of a single gaze has the lowest weight in the visual workload assessment system. Different driver visual workloads were observed depending on the environment of the interchange-merging areas, and based on these, recommendations are proposed to decrease drivers’ workload, thereby increasing road safety.

## 1. Introduction

With the continuous development of domestic freeway networks, their mileage and density are increasing, leading to an increasing demand for interchanges. Because primary facilities connect freeways and other roads of varying levels, interchanges significantly affect a network’s operational efficiency and safety. As the number of interchanges increases, the spacing between them decreases, resulting in high-density clusters. Higher traffic flow leads to increased conflicts in merging areas, and drivers must observe more traffic information, potentially leading to an excessive visual workload and increased accident risk. Therefore, researching drivers’ visual workloads in the merging areas of high-density interchange clusters, revealing the variations and influencing factors in driving workload, and proposing targeted improvement measures are important measures for enhancing driver safety.

In a study on driving workload, Namgung and Sung [1] analyzed the visual data of drivers on circular interchange ramps and found that adopting deceleration measures upon entering circular paths led to changes in the visual search patterns of drivers. Yuan et al. [2] analyzed driver workloads in urban interchange ramp areas and discovered that different types of urban road sections impact driver workload. Xu et al. [3,4] analyzed the differences and influencing factors of driving workload under various interchange clearance conditions and found that drivers exhibited an increasing mental workload at the merging and diverging nose positions of interchanges. Yang et al. [5] employed heart rate to assess driver mental workload at different types of interchange entrances and exits and found that the psychological workload is greater at exit ramps than at entrance ramps. Fallahi et al. [6] studied the influence of traffic flow density on drivers’ mental workloads based on drivers’ physiological indicators and subjective driving workload evaluation scale data and found that drivers’ mental workloads at high traffic flow densities were greater than those at low traffic flow densities. Hu et al. [7] conducted real-vehicle experiments on freeways and established a driving workload model to reveal the impact of merging vehicles, which generate significant risks, on freeway drivers’ workloads and performance. Szychowska M et al. [8] found that when the visual load is small, the stimulation intensity of the driver’s visual information is not high enough, which can produce visual fatigue and lead to traffic accidents. Marquart et al. [9] collected eye-movement data from drivers and analyzed driving workload under different conditions. They found that an increase in driving workload manifested as increases in blink latency, the Perclos principle, gaze duration, pupil dilation, and the index of cognitive activity, as well as decreases in blink duration and gaze variability.

In the realm of visual workload, numerous studies have been conducted on visual workload. Yang and Chen [10] collected indicators such as gaze duration, blink frequency, and heart rate through real-vehicle experiments to reveal the impact mechanism of the road’s spatial environment on driver psychological workload and visual parameters. Faure et al. [11] simulated driving environments of varying complexity to investigate visual indicator patterns and found that blinking behavior can represent the visual workload to a certain extent. Bai et al. [12] found that under different road conditions, the driver’s eye movement dispersion and visual load level will change. The top three elements of the main source of visual load are bicycle users, roadside signposts, and road conditions. Meng et al. [13] gathered data on subjects’ pupil area change rates and heart rate variability and determined the characteristics of drivers’ visual and physiological workloads in different road domain spaces on mountainous two-lane freeways. Gao et al. [14] collected the dynamic visual parameters of drivers in long downhill sections, established a visual workload intensity model, and found that the combination of gentle slopes did not alleviate the drivers’ visual workloads. Clark [15] categorized the influences on driver visual attention as internal and external and evaluated driver visual attention accordingly. Doori et al. [16] studied visual information processing difficulty as a representation of visual workload and found that as vehicle speed increased, the driver’s field of vision narrowed, and the difficulty in processing visual information increased. Xu et al. [17] found that with the change in traffic environment, each visual search index shows significant statistical differences. Especially in the case of roadside risk, the driver’s visual search workload increased significantly. Based on the eye-movement characteristics of drivers, Arakawa [18] used a time-series analysis of drivers’ visual features and established a driver state detection model. Yong-Gang Wang et al. [19] pointed out that when the connecting section of a tunnel group is short, drivers will experience frequent ‘black and white hole effect’, which will produce a large visual load, which is not conducive to the driving safety of this section. Xu et al. [20] found that there are some problems affecting traffic safety in high-density interchange groups, such as a high traffic flow conflict rate, a heavy driver visual load, and a high driving operation pressure. Sun et al. [21] found that drivers’ frequent pupil changes during driving in high-density interchanges are detrimental to their visual cognitive process; drivers’ visual load comprehensive scores in the diversion section of a small clear distance interchange are higher.

Although previous studies have accomplished much in the study of visual workload, few studies on the visual workloads of drivers in high-density freeway interchange-merging areas have been conducted. Furthermore, explorations of visual workload impact indicators and quantitative analyses following visual workload modeling are lacking. Therefore, this study focuses on urban freeway interchange-merging areas by collecting eye-movement data from drivers in various merging areas of a high-density interchange cluster through real-vehicle experiments. Using the gaze time ratio, gaze duration, horizontal search breadth, blink frequency, scanning frequency, and pupil area change rate as evaluation indicators and basing the analysis on the CRITIC weight method, a comprehensive evaluation of the visual workload in interchange-merging areas under different vehicle operating conditions is conducted. Corresponding improvement measures are proposed. The research results provide data support and a theoretical foundation for enhancing the safety of drivers in high-density interchange-merging areas and the design of merging area on-ramps and road traffic capacity.

## 2. Experimental Design

### 2.1. Experimental Scenario

Through satellite map observations and on-site inspections, five interchanges on the inner ring freeway in Chongqing were selected, namely the Cuntan, Wugui, Wutong, Donghuan, and Renhe Interchanges, which form a cluster of high-density interchanges (Figure 1a). Taking the interchange spacing as the discriminant index, interchange groups with more than three interchanges within 10 km on a continuous road are defined as high-density interchange groups [22]. The clear distance between the Wutong, Wugui, and Donghuan Interchanges is less than 1000 m, which does not meet the minimum requirement for interchange spacing stipulated in the “Urban Road Alignment Design Specifications” (CJJ193-2012) [23] and the “Freeway Alignment Design Specifications” (JTGD20-2017) [24]. Consequently, the road sections between these three interchanges are designated as regions with small interchange spacing. An interchange that meets the minimum clear distance requirement is defined as a ‘normal-spacing interchange’, and an interchange that connects small-spacing interchanges in the form of auxiliary lanes or distribution lanes is defined as a ‘composite interchange’. The Wugui Interchange is a composite interchange, whereas the others are normal-spacing interchanges. The basic conditions of these interchanges are shown in Figure 1b and Table 1.

### 2.2. Test Vehicle and Facilities

A Tobii Glasses 2 portable eye tracker (Stockholm, Sweden) was used to collect drivers’ eye-movement data. This device operates at a sampling frequency of 50 Hz and captures videos at a resolution of 1920 × 1080 pixels. It employs dual-image sensors and combined tracking technology for both bright and dark pupils to capture ocular images. The eye tracker ascertains the eye position and gaze point using a dynamic light-emitting diode eye image processing system and a 3D model to gather the driver’s eye-movement data and pupil diameter. A Buick GL8 was chosen as the test vehicle because of its interior space that can accommodate instrumentation and test personnel and offers a high level of driving comfort, as shown in Figure 2.

### 2.3. Participants

A total of 47 drivers participated in the experiment (35 men and 12 women). The participants’ driving experience ranged from 2 to 25 years (M = 11 years). The mileage varied from 20,000 to 500,000 km (M = 140,000 km). Their ages ranged from 25 to 51 years (M = 37 years). Because an eye tracker was used and due to the nature of this experiment, participants were required to have corrected vision above 5.0, no existing vision issues, good driving habits, and experienced no major traffic accidents.

All methods were performed in accordance with the relevant guidelines and regulations. All experimental protocols were approved by the appointing authority or licensing board. Informed consent was obtained from all subjects or their legal guardians.

### 2.4. Experimental Procedure and Data Processing

At the test site, in the debugging stage of the device, WiFi was needed to search the device and then run the data acquisition software (Tobii Pro Glasses Controller 2). Each driver donned an eye tracker, used a hand-held calibration card to visually calibrate the tracker based on the center of the card, and clicked on the calibration in the data acquisition software (Tobii Pro Glasses Controller 2). At the same time, the experimenter informed each driver of the driving route. The drivers then followed their usual driving habits to complete two to three laps along the test route before returning to the starting point, which marked the end of the experiment. After the experiment, all collected data were imported into the Ergo LAB human–machine environmental synchronization platform for preprocessing. Data with an eye-tracking accuracy below 90% were excluded to ensure precision. Using the Ergo LAB platform, the visual segments of drivers in different driving phases in the interchange-merging area were categorized to obtain multiple eye-tracking evaluation indicators for each driving phase, such as gaze duration, gaze point trajectory distribution, scanning amplitude, and pupillary changes.

Anomalous pupil diameter values were observed in the eye-tracking data during the blinking moments of the drivers (Figure 3a). Therefore, before analyzing variations in pupil size, it was necessary to use MATLAB 2019 software to eliminate outlier data and apply filtering to result in corrected pupil diameter data (Figure 3b).

## 3. Modeling Basis

### 3.1. Visual Workload Evaluation Indicators

Drivers perceive the external environment through gaze and scanning behaviors [25]. Visual workload indicators include four main categories, namely gaze, scan, blink, and pupil area, which further comprise gaze duration, frequency, and point position and are categorized into scanning amplitude, frequency, and speed. The blink frequency and pupil area variation rate can intuitively represent the driver’s visual workload. Other indicators are used to characterize the visual workload owing to the difficulty in obtaining visual information [26]. The following six indicators were selected as the evaluation indicators for the visual workload assessment model of drivers, constituting the visual workload evaluation system. Table 2 presents the details of each evaluation indicator, and Figure 4 shows the original data for each indicator.

Indicators 1 and 2 were used to characterize drivers’ gaze behavior. The gaze time ratio signifies the sum of the time spent on gaze points during driving relative to the total time spent on all eye-movement behaviors. Under complex road conditions, drivers allocate more gaze time to extracting road information. The average single-gaze time refers to the time taken for gaze behaviors, which can reflect the complexity of the gaze target and drivers’ difficulty in extracting information. A longer average duration suggests higher road complexity and greater cognitive difficulty in processing information.

Indicators 3 and 4 were designed to characterize driver scanning behavior. The average horizontal search breadth refers to the extent of the driver’s gaze range, with a broader breadth indicating a wider distribution of road information. The average scan frequency reflects the driver’s need for information collection while driving, and a higher scan frequency typically corresponds to the processing of more complex information.

Indicator 5, the blink frequency, reveals the driver’s fatigue level and reflects the driver’s attention and concentration level. A higher blink frequency indicates a more tired or relaxed state, whereas a lower frequency suggests a higher concentration level and the processing of complex road information. Indicator 6, the pupil area change rate, refers to the rate at which the pupil area changes over time and varies across different driving contexts. The pupil area change rate can be used to assess a driver’s level of stress, fatigue, and difficulty in processing road information.

### 3.2. Modeling Approach Based on the CRITIC Weighting Method

The commonly used comprehensive evaluation methods include the analytic hierarchy process (AHP), the entropy method, the CRITIC weighting method, and principal component analysis. The calculation principles and applicability of these methods are shown in Table 3.

When evaluating the visual workload of the drivers, the selected indicator data were interrelated to an extent. The CRITIC method of objective weighting considers the data size and the conflict between the indicators. Hence, the CRITIC weighting method was employed for a comprehensive evaluation of drivers’ visual workloads in freeway interchange-merging areas. Regarding the application of the CRITIC weighting method, Pan et al. [27] compared three common setting forms in conventional intersections and unconventional intersections. From the perspectives of traffic efficiency and the impact on the environment, parking times, delays, and carbon dioxide emissions were selected as evaluation indicators. The CRITIC method was used to calculate the weight of these indicators, and finally the applicable conditions of the four intersection forms were obtained. Therefore, the CRITIC weighting method is also applicable to other urban intersections.

This method ascertains the objective weights of various indicators based on their comparative intensities and conflicts. The variability among indicators is expressed by the standard deviation, with a larger standard deviation indicating greater data volatility and more information, which corresponds to a higher weight. The conflict between indicators is denoted by the correlation coefficient, with a larger correlation coefficient suggesting higher information redundancy, lower conflict, and, consequently, a lower weight.

(1) Establishing the decision matrix. The evaluative sample, denoted as m, consisted of drivers on the road section within the freeway interchange-merging areas. Each evaluative sample comprised n evaluative indicators that reflected the driver’s gaze behavior, scanning behavior, blinking behavior, and pupil change, all of which are eye-movement parameters produced during driving. Letting *X_mn_* represent the nth evaluative indicator of the *m*-th evaluative sample, the original data matrix is as follows:(1)$$X=\left(\begin{array}{c} X_{11}\;\;X_{12}\;\cdots X_{1n}\\ X_{21}\;\;X_{22}\;\cdots X_{2n}\\ \;\vdots \;\;\;\;\;\;\;\;\vdots \;\;\;\;\;\;\;\;\;\;\;\vdots \;\\ X_{m1}\;\;X_{m2}\;\ldots X_{mn} \end{array}\right).$$

(2) Dimensionless matrix processing. Because the evaluative indicators have different units of measurement, normalization was used to eliminate the effects of varying dimensions. A positive indicator formula was applied when larger indicator values correspond to better scores, whereas a negative indicator formula was used when smaller indicator values correspond to better scores. The calculation formulae are as follows:

Positive indicator normalization formula:(2)$$x_{ij}^{\prime }=\frac{x_{ij}-\underset{1\leq j\leq n}{\min}\left\{x_{j}\right\}}{\underset{1\leq j\leq n}{\max}\left\{x_{j}\right\}-\underset{1\leq j\leq n}{\min}\left\{x_{j}\right\}},i=1,2,\cdots ,m.$$

Reverse indicator normalization formula:(3)$$x_{ij}^{\prime }=\frac{\underset{1\leq j\leq n}{\max}\left\{x_{j}\right\}-x_{ij}}{\underset{1\leq j\leq n}{\max}\left\{x_{j}\right\}-\underset{1\leq j\leq n}{\min}\left\{x_{j}\right\}},i=1,2,\cdots ,m,$$
where $x_{ij}$ represents the original data of the matrix and $x_{ij}^{\prime }$ represents the data after normalization.

(3) Calculation of indicator variability, conflict, information capacity, and weight.

Indicator variability:(4)$$\sigma_{j}=\sqrt{\frac{\sum_{i=1}^{m}{\left(x_{ij}^{\prime }-\frac{1}{m}\sum_{i=1}^{m}x_{ij}^{\prime }\right)}^{2}}{m-1}},j=1,2,\cdots ,n.$$

Conflict:(5)$$r_{\alpha \beta }=\frac{\sum_{i=1}^{m}\left(x_{i\alpha }^{\prime }-{\bar{x}}_{\alpha }^{\prime }\right)\left(x_{i\beta }^{\prime }-{\bar{x}}_{\beta }^{\prime }\right)}{\sqrt{\sum_{i=1}^{m}{\left(x_{i\alpha }^{\prime }-{\bar{x}}_{\alpha }^{\prime }\right)}^{2}{\left(x_{i\beta }^{\prime }-{\bar{x}}_{\beta }^{\prime }\right)}^{2}}},$$
(6)$$f_{j}=\sum_{\alpha =1}^{n}\left(1-r_{\alpha j}\right).$$

Information-carrying capacity:(7)$$c_{j}=\sigma_{j}*f_{j}.$$

Weight:(8)$$w_{j}=\frac{c_{j}}{\sum_{j=1}^{n}c_{j}},$$
where $\sigma_{j}$ is the standard deviation of the j-th indicator, $r_{\alpha \beta }$ is the correlation coefficient between the α-th and β-th indicators, ${\bar{x}}_{\alpha }^{\prime }$ and ${\bar{x}}_{\beta }^{\prime }$ are the mean values of the α-th and β-th indicators, $f_{j}$ is the conflict value of the j-th evaluation indicator, $r_{\alpha j}$ is the correlation coefficient between the α-th and *j*-th evaluation indicators, $c_{j}$ is the information-carrying capacity of the *j*-th evaluation indicator, and $w_{j}$ is the weight of the *j*-th evaluation indicator.

Pearson’s method was used to compute the correlation coefficients between the indicators. Based on these results, the calculation of indicator conflictivity was followed by the computation of the information-carrying capacity. The larger the $c_{j}$, the more significant the role of the *j*-th evaluation indicator in the entire evaluation system and, consequently, the greater the weight assigned to that indicator.

(4) Determining the comprehensive evaluation score and evaluation criteria.
(9)$$s_{i}=\sum_{j=1}^{n}w_{j}x_{ij},i=1,2,\cdots ,m,$$
where $s_{i}$ denotes the comprehensive evaluation score for the visual workload of the *i*-th assessment sample.

The comprehensive evaluation score obtained using the CRITIC objective weighting method fell within the [0, 1] interval. The closer the driver’s comprehensive evaluation score for visual workload to 1, the greater the visual workload. The closer the score to 0, the lesser the visual workload and the better the visibility of the road section.

### 3.3. Evaluation Indicator Weight Analysis

The participants differed in their individual characteristics and driving habits, which resulted in variability in the evaluation of the indicator values, information-carrying capacity, and weights during driving. Therefore, using the steps of the CRITIC weighting method, the information-carrying capacity and weights of the six visual evaluation indicators were computed for all drivers when passing through the interchange cluster merging areas (Figure 5). These indicators were used to analyze drivers’ visual workloads in interchange cluster merging areas.

Figure 5a shows the variances in the amount of information contained by the six indicators within the visual workload evaluation system for the interchange-merging areas. The mean average scan frequency (Indicator 4) and mean pupil area change rate (Indicator 6) had the highest proportions, with the average information content values reaching 1.371 and 1.320, respectively. Thus, these indicators captured the greatest amount of information and played the most significant role in the entire evaluation system. These indicators represent the average horizontal search breadth (Indicator 3), with the mean gaze duration having the lowest average information content value of 1.073, indicating that gaze duration has less impact on the evaluation system while driving.

Figure 5b shows that the visual evaluation indicators that capture a higher amount of roadway information carry greater weight, whereas those with lower information content carry less weight. During driving, frequent fluctuations in the gaze frequency and pupil area change rate increase the driver’s visual workload, thereby reducing drivers’ visibility in interchange-merging areas and affecting their driving judgment.

## 4. Results

While driving in the interchange-merging areas, drivers adjust their driving according to different paths, resulting in two types of conditions: (1) transitioning from a ramp to a mainline, where drivers enter a freeway via an interchange ramp and, while accelerating, must observe the traffic conditions on the mainline close to the ramp side until they successfully merge into the interchange; and (2) the mainline driving situation, where vehicles continuously traveling on the mainline intersect with vehicles entering from ramps, requiring drivers to monitor vehicles merging from the ramp side and decreasing the speed for the merge. Therefore, a visual workload evaluation was conducted for drivers in both ramp-to-mainline and mainline driving conditions, and analyses were conducted on the drivers’ visual workloads across different driving sections of the interchange-merging areas, considering the presence of obstacles, the number of ramp lanes, and the interchange-merging form.

### 4.1. Evaluation of Visual Workload for the Ramp-to-Mainline Driving Condition

Figure 6a shows the driving condition from the ramp to the mainline, and Figure 6b shows the ramp section, which refers to the stretch from the start of the ramp to the merge nose end, which was measured to be 150 m. The merging section refers to the stretch from the merge nose end to the end of the acceleration lane. Using the CRITIC weighting method, the visual workload comprehensive evaluation scores for drivers at each interchange junction for the ramp-to-mainline condition were calculated to obtain the distribution of the drivers’ visual workloads for different conditions in the ramp and merging sections (Figure 7).

Figure 7a shows the visual workload comprehensive score graph of the ramp section: the comprehensive evaluation scores of the drivers’ visual workloads in the ramp section with obstructions are generally higher than those observed in the unobstructed conventional interchange. Thus, when obstacles obstruct the view in the interchange-merging area, the visual demand on drivers increases; the driver is unable to observe vehicles traveling on the main roadway, and the driver attempts to obtain effective information by increasing their scanning frequency and visual workload.

The visual workload decreases in the order of single-lane, three-lane, and two-lane ramps. The Donghuan Interchange’s No. 2 ramp (DH2ZD), which has three lanes, has the highest average comprehensive visual workload score, whereas the Wugui Interchange’s No. 1 ramp (WG1ZD) has the lowest. Vehicles merging from the single-lane Renhe Interchange No. 2 ramp (RH2ZD) onto the mainline face complex traffic conditions because of the large number of vehicles on the mainline near the ramp and the short acceleration lane; some vehicles cannot successfully merge—drivers must wait for gaps to enter the mainline at the merged nose end. Hence, they encounter a greater visual workload there than on a two-lane ramp. Additionally, drivers on a three-lane ramp face even more vehicles and complex road conditions, requiring the processing of more vehicle and road information, which leads to a higher driving workload compared with a two-lane ramp.

In merging areas, the acceleration lanes run parallel to the mainline and gradually connect to it. Vehicles can accelerate in the acceleration lane and then merge into the mainline. However, in the direct type, acceleration lanes connect directly to the mainline, allowing vehicles to merge from the ramp straight into traffic. Figure 7b shows that the lengths of the acceleration lanes increased from CT1HL to WG1HL and from RH2HL to WT1HL. These results indicate that the drivers’ comprehensive visual workload scores decreased with the increasing length of the acceleration lanes. The longer the acceleration lane, the more time the drivers have to process information and the better the visibility, which is beneficial for reducing the driver’s visual workload. The average comprehensive visual workload score for the Donghe Interchange No. 1 merging section (DH1HL) is the highest because the merging sections of the other interchanges have four lanes and the mainline into which it flows has five lanes. This results in a higher traffic volume and more information for drivers to observe. Hence, a higher comprehensive visual workload score is obtained at this interchange.

Figure 8 presents a comparative analysis of the visual workloads of drivers at the interchange ramps and merging segments when the vehicles travel from the ramp to the mainline. It can be seen that in addition to CT1 and DH1, the visual workloads at the merging segments are generally greater than or equal to those at the ramp segments. Because the intersection of the two roads in the merging section is more complicated than that in the ramp section, the traffic information is more complicated. In these environments, drivers must concentrate more as they are required to accelerate and change lanes while observing the vehicles traveling on the mainline. Consequently, a comprehensive assessment of the drivers’ visual workload is typically higher at merging segments than at ramps. The ramp section of CT1 is blocked by traffic facilities or greening facilities, while the other interchange ramps in this figure are not blocked by obstacles. The blocking of greening facilities leads to the visual load of the ramp section being greater than that of the confluence section. The confluence section of DH1 is five lanes wide. Because the confluence section of other interchanges is four lanes wide, the traffic volume is large. When the vehicle merges with the main line, the vehicle interweaves frequently, which can cause traffic conflict. Therefore, the driver’s visual load is high when driving on this section.

### 4.2. Assessment of Visual Workload under Mainline Driving Conditions

Figure 9a shows the driving conditions of vehicles traveling on the mainline. Regarding the mainline driving conditions (Figure 9b), ZX represents Mainline Section I, which refers to the section where drivers travel on the mainline to the merging nose end, which was measured to be 150 m. ZX represents Mainline Section II, referring to the section where drivers who are adjacent to the ramp lanes continue to drive along the mainline, past the merging nose end.

The comprehensive visual workload scores for each driver at the Cuntan Interchange No. 2, Wutong Interchange No. 2, and Renhe Interchange No. 1 mainline driving conditions were calculated (Figure 10). A comparison of Mainline Sections I and II revealed that the visual workloads for Section I at the Wutong Interchange No. 2 and Section I at the Renhe Interchange No. 1 are higher than those for Mainline Section II. This is because drivers in Mainline Section I must pay attention to whether there will be a ramp vehicle in front, which increases the amount of information to process, thereby increasing the visual workload. By contrast, at Cuntan Interchange No. 2, the visual workload for Mainline Section II is higher than that for Mainline Section I. This can be attributed to a traffic signal light ahead of the merging nose end at Cuntan Interchange No. 2, which causes drivers to divert some of their attention to the signal changes. Visual scanning frequency is influenced by the signal light and queued vehicles, thereby increasing the visual workload for Mainline Section II compared with Mainline Section I.

Among all mainline driving conditions, the highest comprehensive visual workload score was observed for Section I at the Wutong Interchange No. 2. Here, a high volume of traffic merges from the ramps, and the traffic volume on the mainline section itself is substantial. Frequently, mainline vehicles must reduce their speed and change lanes to accommodate vehicles from ramps, leading to increased traffic conflicts in the merging area and higher visual workloads for drivers.

## 5. Measures to Enhance Safety in Interchange-Merging Areas

A comprehensive visual workload assessment revealed that drivers receive complex road traffic information when they travel through interchange-merging areas. The obstruction conditions, structural configuration, and acceleration lane design in the merging areas affect the driver’s visual characteristics. Consequently, the following improvements are suggested.

### 5.1. Improve Visibility in Merging Areas

When vehicles drive in ramp-triangle areas with obstacles (Figure 11a), drivers have difficulties observing the vehicles driving on the mainline and at the entrance merge nose point, leading to decreased driver visibility recognition and shorter reaction times. When sudden circumstances arise, it is challenging to act quickly. Our research has found that drivers experience a higher visual workload when driving through merging areas with obstructions compared with those without obstructions. Therefore, merging areas without obstructions are more conducive for drivers to observe traffic conditions at the mainline entrance, process traffic information in advance, and establish a prejudgment of the merging process. Therefore, landscaping in the merging area should be replaced with shrubbery that does not impede driver visibility (Figure 11b) to satisfy the sight distance requirements of the visibility triangle in the merging area (Figure 11c).

### 5.2. Adjust the Configurations of Merging Areas

The number of acceleration lanes in the interchange-merging ramp section affects the visual characteristics of drivers. According to calculations from the visual workload model, the comprehensive visual workload score when merging into the mainline is higher for three lanes than for two lanes. Drivers process a greater amount of traffic information and have a higher visual workload when traveling in three lanes (Figure 12a). Therefore, provided that the requirements for the ramp’s traffic capacity are satisfied, three-lane ramps should be transitioned into two-lane ramps, starting 150 m from the merging nose end on the ramp section, and conspicuous signs in the triangular area should be installed (Figure 12b). Thus, information complexity in the merging area is reduced, addressing challenges to rapid merging due to the high volume of information in a three-lane ramp merging section.

### 5.3. Optimize the Design of Acceleration Lanes in Merging Areas

Drivers must accelerate in the acceleration lane when merging from a ramp to a mainline in an interchange-merging area to reduce the speed differential between the ramp and mainline vehicles, thereby minimizing traffic conflicts. This research shows that interchanges between a single-lane ramp merging section and direct acceleration lanes result in greater visual workloads for drivers (Figure 13a). Therefore, the dashed portion of the guiding line should be converted into a solid line at a distance based on the acceleration lane’s length to extend the acceleration time and reduce the drivers’ visual workload in the interchange-merging area (Figure 13b). Furthermore, connecting single-lane ramps to short direct acceleration lanes should be avoided (Figure 13c). An acceleration lane extension can help to reduce the number of vehicles queuing on the ramp and waiting to merge into the mainline, and thus decrease interference with mainline vehicles (Figure 13d).

## 6. Discussion

Previous studies on visual workloads have focused on drivers’ mental workloads in tunnels or on freeways [11,28] or explored a single visual indicator. Few studies have been conducted on the visual workload impact indicators of high-density interchanges, and explorations of driver eye-movement data and quantitative analysis after modeling are lacking. In this study, a real-vehicle driving test was conducted with 47 participants in five interchanges of the Chongqing Inner Ring Freeway, involving three small-spacing, one composite, and one normal-spacing interchange. The proportions of fixation time, fixation duration, horizontal search breadth, blink frequency, scanning frequency, and pupil area change rate were selected to analyze the driver visual characteristics and evaluate the visual workloads for the two conditions in the merging areas.

Xiaoping et al. [29] collected drivers’ eye-movement data under different fatigue states and found that an increase in fatigue leads to an increase in blink time and decreases in scanning time and pupil area. This study found that the blink frequency and pupil area change rate also reflect the driver’s concentration and the difficulty experienced while processing road-section information. Benedetto et al. [30] found that the driving task size affects a driver’s blink frequency and duration. In this study, it was found that when drivers accelerate and change lanes in a merging section, they change their blinking frequencies, have higher visual workloads, and carefully observe the vehicles traveling on the mainline. Thus, drivers’ visual workloads in the merging section are higher than those in the ramp section. Previous studies have explored the changing rules of visual indicators by simulating various driving environments. Blink behavior has been used as a single indicator to characterize visual workload [11]. In this study, the visual workload size was explored for different driving environments, including ramp sections with or without obstructions, different numbers of lanes, and different acceleration lane lengths. In addition, this study not only used blink frequency as an evaluation indicator but also found that eye-movement indicators besides blink frequency, such as fixation time, scanning frequency, and pupil area change rate, can characterize the visual workload size. Specifically, the indicator weights of the scanning frequency and pupil area change rate were the largest, with the greatest impact on drivers’ visual characteristics. Kondyli and Elefteriadou [31] found that a key factor in traffic conflicts is an insufficient length of the acceleration lane, which causes vehicles to enter the mainline at a low speed. This study found that the shorter the acceleration lane, the greater the visual workload, further improving the understanding of drivers’ visual workloads in interchange areas. Driving safety can be improved by transitioning from a three-lane ramp section to a two-lane acceleration lane format and by changing the dashed part at the front end of the guidelines to solid.

Limited by the experimental conditions, this study examined the influence of merging area obstructions, ramp lane numbers, and acceleration lane forms on driver visual characteristics. Future research should explore the effects of road longitudinal slope, ramp radius, traffic flow, lane change, vehicle types, driving conditions, and other factors on driver visual characteristics. This study was conducted in a specific region with certain characteristics. The research findings for mountain cities may differ from those of other cities. If this study could be conducted in other regions as well, the conclusions would be more universal and representative.

## 7. Conclusions

(1)According to the CRITIC weighting method, the average scan frequency and average pupil area change rate had the highest weights, indicating a great impact on driver visual characteristics. The average single-gaze duration had the lowest weight in the visual workload assessment system for the interchange-merging areas, indicating a relatively smaller influence.(2)Drivers typically experience a higher visual workload when traveling on ramps with elements that obscure their view compared with unobstructed ramp sections. An appropriate arrangement of greenery and traffic facilities in triangular ramp areas can provide drivers with an improved visual recognition environment, facilitating the timely acquisition of information regarding traffic conditions in interchange-merging areas. The visual workloads of drivers on ramps with different numbers of lanes were ranked from highest to lowest as follows: single lane > three lanes > two lanes.(3)The longer the acceleration lane in the merging area, the lower the driver’s visual workload. A longer acceleration lane helps drivers to reduce their speed differential with vehicles on the mainline segment. Adopting a parallel acceleration lane design and extending the lane length in the merging section can alleviate the driving workload.(4)In the ramp-to-mainline condition, the driving environment of the merging section was more complex than that of the ramp section, and the drivers’ visual workloads in the merging section were greater than or equal to those in the ramp section. In the mainline condition, the integrated visual workload score from the merging nose point to the merging nose end was higher than that observed from the merging nose end to the end of the acceleration lane. From the merging nose point to the merging nose end, drivers must observe whether vehicles are merging from the front and simultaneously change lanes to the left, which requires them to increase their attention.(5)Based on the evaluation results, targeted improvement measures are proposed: enhancing the visibility distance of sight triangles, removing greenery from these areas, converting greenery to shrubbery, adjusting the geometric configuration of merging areas, reducing the number of lanes from three to two in the merging segment, and improving lane marking design.

## Figures and Tables

**Figure 1 sensors-24-06247-f001:**
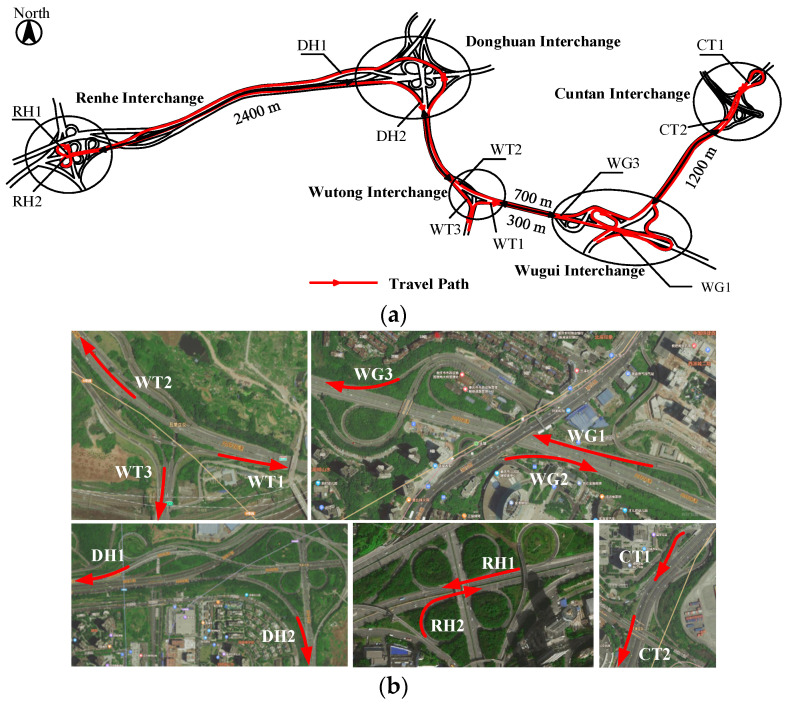
On-site experimental location. (**a**) Test route and location. (**b**) Satellite images of the interchange confluence area.

**Figure 2 sensors-24-06247-f002:**
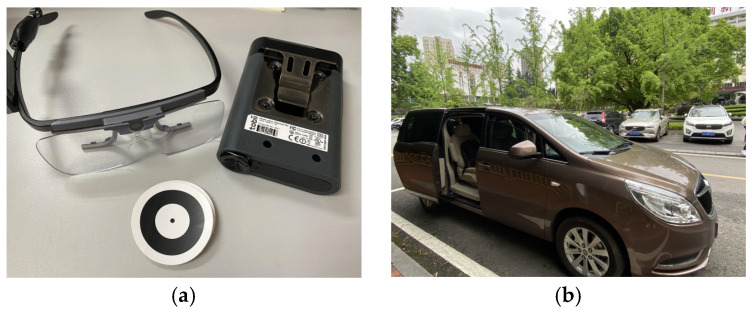
Experimental apparatus and vehicle. (**a**) Eye tracker. (**b**) Test vehicle.

**Figure 3 sensors-24-06247-f003:**
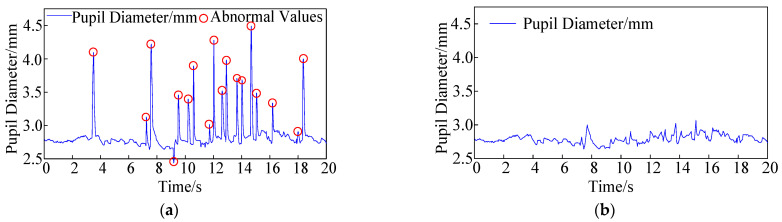
Collected pupil diameter values. (**a**) Pupil diameter before outlier exclusion. (**b**) Pupil diameter after outlier exclusion.

**Figure 4 sensors-24-06247-f004:**
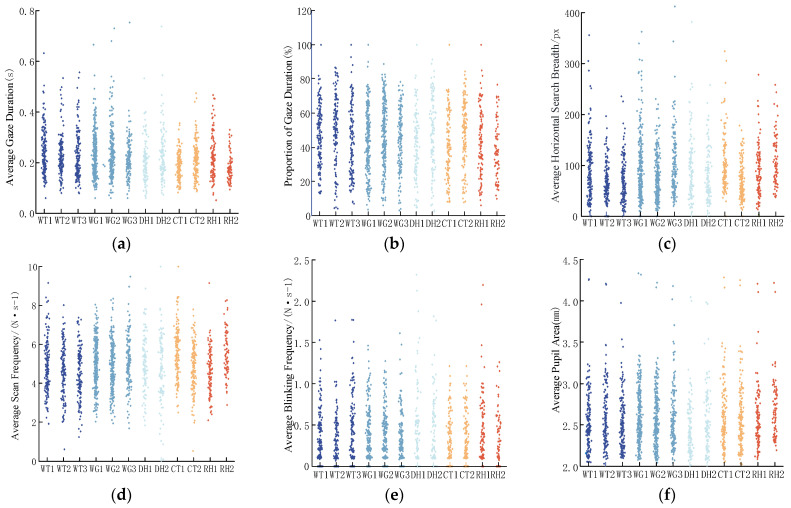
Original data of the indicators. (**a**) Average gaze duration. (**b**) Proportion of gaze duration. (**c**) Average horizontal search breadth. (**d**) Average scan frequency. (**e**) Average blinking frequency. (**f**) Average pupil area.

**Figure 5 sensors-24-06247-f005:**
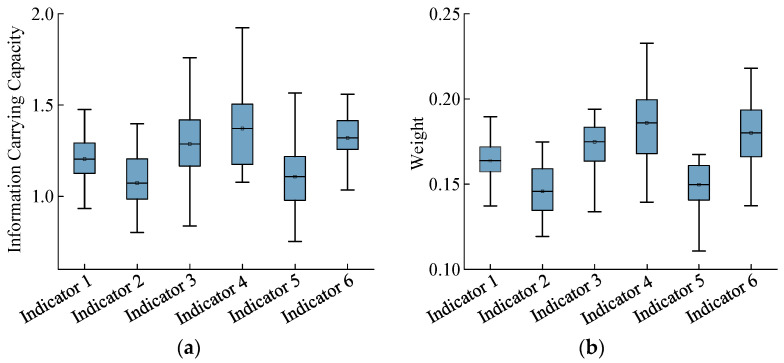
Comparison of indicator values in the CRITIC visual workload assessment system. (**a**) Information capacity indicators. (**b**) Weight indicators.

**Figure 6 sensors-24-06247-f006:**
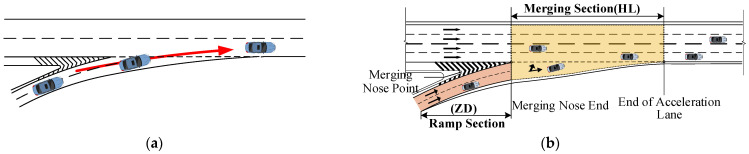
Ramp-to-mainline driving. (**a**) Ramp-to-mainline driving condition route. (**b**) Ramp-to-mainline area segmentation. Red arrow shows the direction.

**Figure 7 sensors-24-06247-f007:**
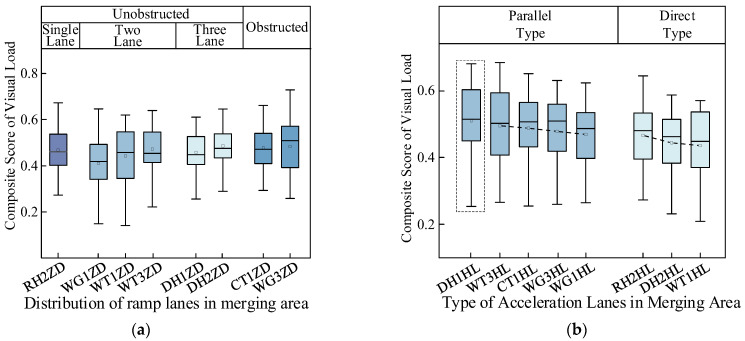
Evaluation results of visual workload on different road sections under the ramp-to-mainline condition. (**a**) On-ramp visual workload comprehensive scores. (**b**) Merging visual workload comprehensive scores.

**Figure 8 sensors-24-06247-f008:**
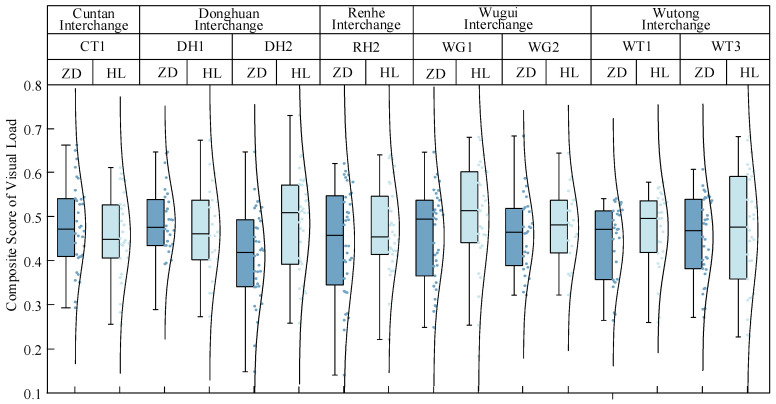
Comparison of visual workloads in ramp and merging sections.

**Figure 9 sensors-24-06247-f009:**
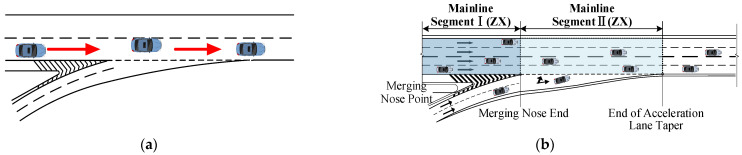
Mainline driving. (**a**) Main driving condition route. (**b**) Mainline area segmentation. Red arrow shows the direction.

**Figure 10 sensors-24-06247-f010:**
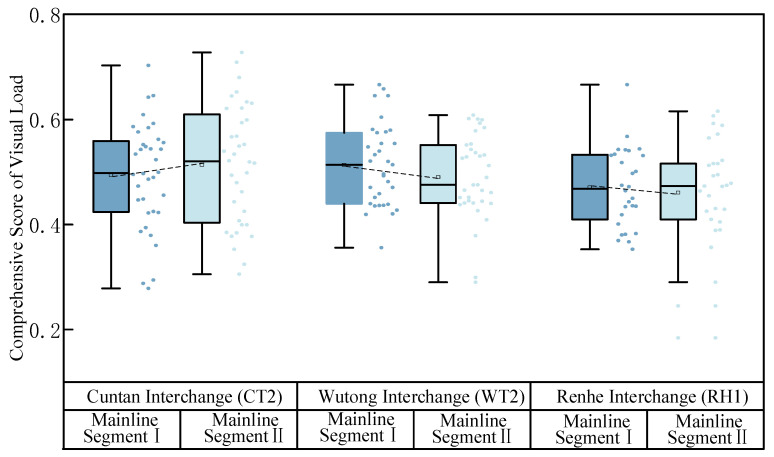
Comprehensive visual workload across different road sections under mainline driving conditions.

**Figure 11 sensors-24-06247-f011:**
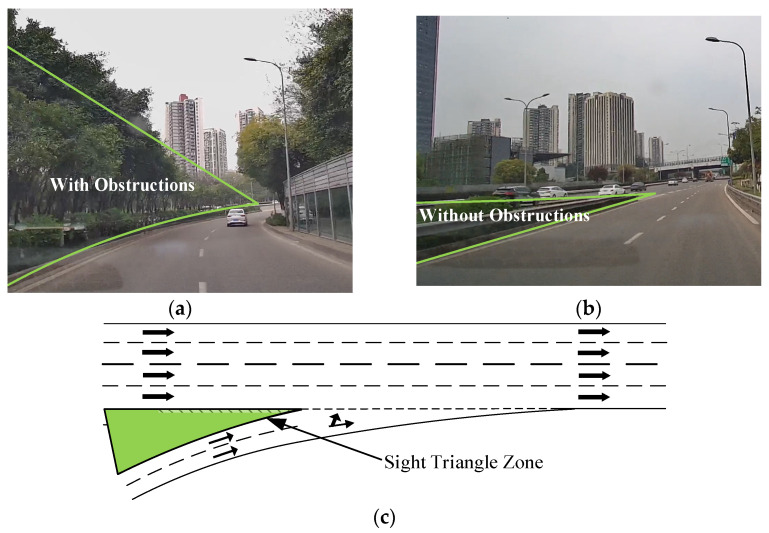
Improvement of sight distance in merge areas. (**a**) Poorer visibility with obstruction. (**b**) Visibility was better without obstructions. (**c**) Schematic of improved visibility in the merged areas.

**Figure 12 sensors-24-06247-f012:**
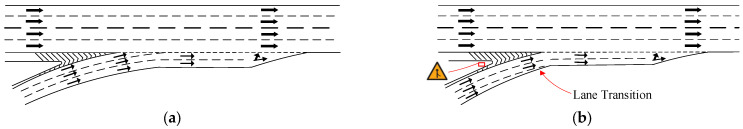
Improvements in the structural design of merge areas. (**a**) Ramp section with a three-lane acceleration lane. (**b**) Ramp section transitioning from three to two lanes, plus an acceleration lane.

**Figure 13 sensors-24-06247-f013:**
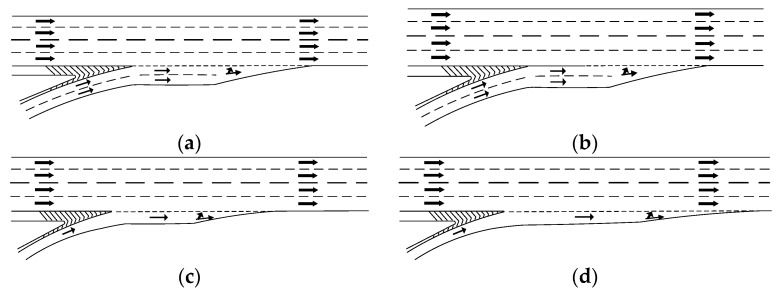
Suggestions for improvement measures for acceleration lanes. (**a**) The leading edge of the guideline is dashed (before change). (**b**) The dashed part at the front end of the guideline is changed to solid (after change). (**c**) The acceleration lane is shorter (before change). (**d**) The acceleration lane length is extended (after change).

**Table 1 sensors-24-06247-t001:** Basic information about the test interchange.

Interchange Name	Confluence Area ID	Vehicle Operating Condition	Acceleration Lane Type	Interchange Spacing Type
Wutong Interchange	WT1	Ramp → Mainline	Direct	Small Spacing
	WT2	Mainline Driving	Parallel	Small Spacing
	WT3	Ramp → Mainline	Parallel	Normal Spacing
Wugui Interchange	WG1	Ramp → Mainline	Parallel	Composite
	WG2	Ramp → Mainline	Parallel	Normal Spacing
	WG3	Ramp → Mainline	Parallel	Small Spacing
Donghuan Interchange	DH1	Ramp → Mainline	Parallel	Normal Spacing
	DH2	Ramp → Mainline	Direct	Small Spacing
Cuntan Interchange	CT1	Ramp → Mainline	Parallel	Normal Spacing
	CT2	Mainline Driving	Direct	Normal Spacing
Renhe Interchange	RH1	Mainline Driving	Direct	Normal Spacing
	RH2	Ramp → Mainline	Direct	Normal Spacing

**Table 2 sensors-24-06247-t002:** Driver visual workload assessment indicator.

Serial No.	Indicator	Unit	Description
Indicator 1	Proportion of Gaze Duration	%	Degree of information processing activity
Indicator 2	Average Single Gaze Duration	ms	Difficulty of information acquisition
Indicator 3	Average Horizontal Search Breadth	px	Distribution of visual information search area
Indicator 4	Average Scan Frequency	N·s^−1^	Demand for receiving and recognizing information
Indicator 5	Average Blink Frequency	N·s^−1^	Degree of driver fatigue and psychological workload
Indicator 6	Average Pupil Area Change Rate	%	Degree of driver’s psychological workload

**Table 3 sensors-24-06247-t003:** The principle and applicability of different weight calculation methods.

Evaluation Method	Principle	Merits and Demerits	Serviceability
AHP	The weight is calculated by using the relative information size of the number.	It can deal with the combination of qualitative and quantitative problems; expert scoring empowerment has certain subjectivity.	Calculate the weight of the index of the multi-level structure
Entropy method	Using the amount of information to calculate the weight	Objective empowerment, to avoid human factors deviation; it has great dependence on samples, and the weight is determined only by data fluctuation.	There are many indexes, and the bottom index calculates the weight.
CRITIC weighting method	The correlation between data fluctuations and indicators is used for calculation.	Considering the amount of information while taking into account the correlation between indicators.	There is a certain correlation between the index data.
Principal component analysis	Information concentration, data dimension reduction	The calculation is simple; when the factor symbol of the principal component is positive or negative, the meaning of the evaluation function is not clear.	When there are many indexes, the principal component weight can be obtained by dimension reduction.
Factor analysis method	Information concentration, data dimension reduction	Factor variables are more interpretable; the least square method used to calculate the factor score may fail.	When there are many indexes, the interpretable factor weights are extracted.

## Data Availability

The datasets generated during this study are available from the corresponding author upon request.

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
