# Peer review of "Study on the Driver Visual Workload in High-Density Interchange-Merging Areas Based on a Field Driving Test"

_sensors, 2024, doi:10.3390/s24196247_

Round 1

Reviewer 1 Report

Comments and Suggestions for Authors

The manuscript provides a comprehensive assessment of visual workload in interchange-merging areas, utilizing real-vehicle driving tests and various visual indicators such as fixation time, blink frequency, and pupil area change rate. The detailed analysis of visual workload under different driving conditions, including ramp-to-mainline and mainline scenarios, is a valuable contribution to understanding driver behavior and safety in complex traffic environments. The study's focus on evaluating and improving visual workload through practical measures is noteworthy. However, to enhance the quality and impact of the paper, several issues need to be addressed:

1.There are many details in the text that need to be carefully checked by the author, such as the correctness of the title format, the use of various rank numbers, informal terms, ambiguous/difficult to read sentences, grammatical errors, and vague terms. If any of the above errors occur, please revise the manuscript completely.

2.The use of the CRITIC weighting method for visual workload assessment is appropriate; however, the paper should provide a more detailed explanation of why this method was chosen over others (such as Principal Component Analysis or Multi-Criteria Decision Analysis). A comparison of these methods would offer deeper insights into the choice of method.

3.While the CRITIC weighting method has proven effective for visual workload assessment, how generalizable is this method across different types of intersections and varying traffic conditions? Is there evidence to suggest that this method is equally effective at intersections in other cities or countries?

4.Does the paper address how the weights of visual indicators may adjust according to dynamic driving conditions (such as changes in traffic flow or road geometry) during real driving? Has consideration been given to how these dynamic adjustments impact visual workload assessment?

5.The references cited in the paper are somewhat outdated. It is recommended to update them with more recent studies to ensure that the paper reflects the latest theoretical and technological advancements.

Comments on the Quality of English Language

There are many details in the text that need to be carefully checked by the author

Author Response

Thank you for your professional review of the manuscript, we are very grateful for your valuable suggestions. We have carefully revised the content of the manuscript.

Comments 1: There are many details in the text that need to be carefully checked by the author, such as the correctness of the title format, the use of various rank numbers, informal terms, ambiguous/difficult to read sentences, grammatical errors, and vague terms. If any of the above errors occur, please revise the manuscript completely.

Response 1: Thanks to the reviewers for their valuable comments, we are sorry for our carelessness. We conducted a comprehensive verification of this article has corrected careless mistakes.

Comments 2:The use of the CRITIC weighting method for visual workload assessment is appropriate; however, the paper should provide a more detailed explanation of why this method was chosen over others (such as Principal Component Analysis or Multi-Criteria Decision Analysis). A comparison of these methods would offer deeper insights into the choice of method.

Response 2: Thanks to the valuable comments made by the reviewers, we have added a comparison table of various comprehensive evaluation methods. After comparing with these methods, we find that CRITIC weighting method is the best method we can choose.

Comments 3:While the CRITIC weighting method has proven effective for visual workload assessment, how generalizable is this method across different types of intersections and varying traffic conditions? Is there evidence to suggest that this method is equally effective at intersections in other cities or countries?

Response 3: We thank you for your advice. We have determined to select the CRITIC weighting method in Chapter 3.2 and cited the literature to illustrate the versatility of the method.

Comments 4:Does the paper address how the weights of visual indicators may adjust according to dynamic driving conditions (such as changes in traffic flow or road geometry) during real driving? Has consideration been given to how these dynamic adjustments impact visual workload assessment?

Response 4: Thank you for your valuable comments. The visual index is the eye movement behavior naturally generated by the driver according to the dynamic changes of different road environments and traffic flows during the actual driving process. The 47 drivers in this paper drove 2-3 laps to obtain real data, and selected 6 indicators. In addition to the proportion of gaze duration, the rest are the average indicators of the driver 's driving in the confluence area ( such as the average single gaze duration ), so the average value of the driver in this interchange confluence area is taken. The average value of each person in each interchange confluence area will also vary due to the difference in the geometry of each interchange confluence area. According to the formula in the 3.2 chapter and the box line diagram in Figure 5 ( b ), the weight of the visual index in this article is generated by aggregating 47 drivers driving on different roads. The weight is calculated according to the value generated by each driver. The amount of information is given a higher weight value.

Comments 5:The references cited in the paper are somewhat outdated. It is recommended to update them with more recent studies to ensure that the paper reflects the latest theoretical and technological advancements.

Response 5: Thank you for your valuable comments. We do find that some literature is outdated and has been updated.

Reviewer 2 Report

Comments and Suggestions for Authors

The manuscript titled "Study on the Driver Visual Workload in High-Density Interchange-Merging Areas Based on a Field Driving Test" presents a well-structured and methodologically sound investigation into a pertinent issue in traffic safety and human factors engineering. The use of real-vehicle driving tests and the CRITIC objective weighting method adds significant value to the research, providing insights that are both practical and theoretically robust. However, there are several areas where the manuscript could be strengthened:

1. The introduction provides a good overview of the context and significance of the study. However, the literature review could be expanded to include more recent studies on visual workload in high-density traffic environments. This would help to better position the research within the current state of the field.

2. The methodology is a lack of detail regarding the calibration process for the Tobii Glasses 2 eye tracker. Since the accuracy of visual workload measurements heavily depends on precise calibration, more information should be provided on how this was achieved and validated. The selection criteria for the five interchanges should be clarified further. Were these chosen randomly, or based on specific traffic or design characteristics? Providing a rationale for the selection of these particular interchanges would strengthen the study’s credibility.

3. The interpretation of the CRITIC weights could benefit from additional discussion. Specifically, why do the average scan frequency and average pupil area change rate have the highest weights? Linking these findings back to theoretical frameworks on visual attention and workload would be beneficial. Additionally, the data on driver gender and age is provided, but there is no analysis on how these factors might influence visual workload. It could be valuable to explore whether there are any significant differences in workload across different demographic groups.

4. The discussion section should more explicitly address the limitations of the study, particularly regarding the generalizability of the findings. The study focuses on a specific geographical area (Chongqing), which may limit the applicability of the results to other regions with different traffic conditions or road designs. The conclusion could be strengthened by suggesting directions for future research. For example, exploring how different vehicle types or driving conditions (e.g., night driving) might affect visual workload in interchange-merging areas would be a useful extension of this work.

5. Figures and tables are well-presented, but Figure 8 could benefit from a more detailed explanation in the text. It’s not immediately clear why some interchanges show higher visual workloads in merging segments compared to ramps, and this warrants further clarification.

6. There are a few typographical errors and awkward phrasings throughout the manuscript. A thorough proofread is recommended to enhance readability. The references section is comprehensive, but it may be helpful to ensure that all cited studies are the most current and relevant to the topic.

Author Response

Thank you for your professional review of the manuscript, we are very grateful for your valuable suggestions. We have carefully revised the content of the manuscript.

Comments 1: The introduction provides a good overview of the context and significance of the study. However, the literature review could be expanded to include more recent studies on visual workload in high-density traffic environments. This would help to better position the research within the current state of the field.

Response 1: We sincerely thank you for your valuable comments. We have carefully examined the literature and have supplemented the newer high-density interchange related literature.

Comments 2:The methodology is a lack of detail regarding the calibration process for the Tobii Glasses 2 eye tracker. Since the accuracy of visual workload measurements heavily depends on precise calibration, more information should be provided on how this was achieved and validated. The selection criteria for the five interchanges should be clarified further. Were these chosen randomly, or based on specific traffic or design characteristics? Providing a rationale for the selection of these particular interchanges would strengthen the study’s credibility.

Response 2: Thank you for your suggestion, we describe the calibration details of the Tobii Glasses 2 eye tracker before use in the 2.4 Experimental Procedure and Data Processing. 2.2. Test Vehicle and Facilities also explains how the Tobii Glasses 2 eye tracker realizes data acquisition. The choice of interchanges is not completely random. First of all, we compare and select the five interchanges through satellite maps and conduct field exploration and research. When choosing, we need some interchanges to meet the relevant conditions of our high-density interchange group, that is, more than three interchanges appear on 10 km continuous roads, and the type of interchange cannot be too single. This part has been supplemented at the 2.1. Experimental Scenario.

Comments 3: The interpretation of the CRITIC weights could benefit from additional discussion. Specifically, why do the average scan frequency and average pupil area change rate have the highest weights? Linking these findings back to theoretical frameworks on visual attention and workload would be beneficial. Additionally, the data on driver gender and age is provided, but there is no analysis on how these factors might influence visual workload. It could be valuable to explore whether there are any significant differences in workload across different demographic groups.

Response 3: Thank you for your valuable advice. This is the real data calculated by taking the real vehicle data of 47 drivers into the formula of the 3.2. Modeling Approach Based on the CRITIC Weighting Method critic weight method, and the data is represented by the box line diagram in Figure 5. Finally, the visual load comprehensive evaluation score is obtained by taking the obtained weight into the formula ( 9 ), which is related to the visual load. In our follow-up work, we will incorporate the gender age you said into our research scope, and carry out detailed research on this. The driver 's gender age provided in this paper is the universality of the personnel who explain this experiment.

Comments 4: The discussion section should more explicitly address the limitations of the study, particularly regarding the generalizability of the findings. The study focuses on a specific geographical area (Chongqing), which may limit the applicability of the results to other regions with different traffic conditions or road designs. The conclusion could be strengthened by suggesting directions for future research. For example, exploring how different vehicle types or driving conditions (e.g., night driving) might affect visual workload in interchange-merging areas would be a useful extension of this work.

Response 4: Thank you for your comments. We have revised the discussion section, and you mentioned that the different vehicle types or driving conditions are included in the research category that we lack, and also said in the discussion section that this study is limited by geographical conditions, lack of universality and other issues.

Comments 5: Figures and tables are well-presented, but Figure 8 could benefit from a more detailed explanation in the text. It’s not immediately clear why some interchanges show higher visual workloads in merging segments compared to ramps, and this warrants further clarification.

Response 5: We thank you for your advice. The following of Figure 8 explains why the visual load of the confluence section is greater than that of the ramp section, because the confluence section intersects with two roads, and the driving information is complex. The conclusion is that the conclusion obtained from a large amount of data is true, and the conclusion of the confluence area with two interchanges is different from this.

Comments 6: There are a few typographical errors and awkward phrasings throughout the manuscript. A thorough proofread is recommended to enhance readability. The references section is comprehensive, but it may be helpful to ensure that all cited studies are the most current and relevant to the topic.

Response 6: Thank you for your careful reading, we are sorry for our carelessness. We conducted a comprehensive verification of this article to correct careless errors and replace outdated references with newer references.

Reviewer 3 Report

Comments and Suggestions for Authors

In this manuscript, the author studies the visual load of the driver when driving in the high-density interchange area, and selects the appropriate indicators to evaluate it, which has certain practical and theoretical significance. The manuscript is generally well written, with clear and understandable statements. There are a few contents that need to be supplemented and corrected.

1. The  literature review includes outdated papers and need to be update.

2. The high-density interchange mentioned in the title is not defined in the manuscript. please make appropriate supplements.

3. The article only explains the small spacing interchange, but does not show the definition of other types of interchange to the reader, and it is suggested to supplement the relevant instructions to clarify the research object.

4. The y-axis format of Figure 4(d) is incorrect and needs correction. The y-axis values in Figure 7(b) are inconsistent with those in (a) and contain formatting errors; please correct them.

5. Figure 8 shows two Renhe interchanges. Please verify and correct this.

6. The average scanning rate and average pupil area mentioned in lines 274 and 278 cannot be clearly correlated with the content presented in Figure 5. Please specify which metrics in Figure 5 correspond to these two indicators.

Comments on the Quality of English Language

Pretty good.

Author Response

Thank you for your professional review of the manuscript, we are very grateful for your valuable suggestions. We have carefully revised the content of the manuscript.

Comments 1: The  literature review includes outdated papers and need to be update.

Response 1: Thank you for your valuable comments. We do find that some of the literature is older and has updated outdated papers.

Comments 2: The high-density interchange mentioned in the title is not defined in the manuscript. please make appropriate supplements.

Response 2: Thank you for your valuable comments. We supplemented the relevant literature of high-density interchange in 2.1, so that readers can understand high-density interchange.

Comments 3: The article only explains the small spacing interchange, but does not show the definition of other types of interchange to the reader, and it is suggested to supplement the relevant instructions to clarify the research object.

Response 3: We thank you for your advice. We define the meaning of other interchanges ( such as : normal interchange, composite interchange ) in 2.1.

Comments 4: The y-axis format of Figure 4(d) is incorrect and needs correction. The y-axis values in Figure 7(b) are inconsistent with those in (a) and contain formatting errors; please correct them.

Response 4: Thanks to the reviewers for their valuable comments, we are sorry for our carelessness. We have corrected the contents of the figure.

Comments 5: Figure 8 shows two Renhe interchanges. Please verify and correct this.

Response 5: Thanks to the reviewers for their valuable comments, we are sorry for our carelessness. The second of them should be Wugui Interchange, we have corrected it.

Comments 6: The average scanning rate and average pupil area mentioned in lines 274 and 278 cannot be clearly correlated with the content presented in Figure 5. Please specify which metrics in Figure 5 correspond to these two indicators.

Response 6: Thank the reviewers for their valuable comments. Since our negligence did not mention the indicators corresponding to the average scanning rate and the average pupil area in the text, it has now been supplemented for readers to read.